# Improving Graph Matching with Positional Reconstruction Encoder-Decoder Network

**Yixiao Zhou**
Wangxuan Institute of Computer Technology
Peking University

**Ruiqi Jia**
Wangxuan Institute of Computer Technology
Peking University

**Hongxiang Lin**
Wangxuan Institute of Computer Technology
Peking University

**Hefeng Quan**
School of computer science and engineering
Nanjing University of Technology

**Yumeng Zhao**
School of Artificial Intelligence
Beijing University of Posts and Telecommunications

**Xiaoqing Lyu**[*]
Wangxuan Institute of Computer Technology, Beijing Institute of Big Data Research
Peking University

## Abstract

Deriving from image matching and understanding, semantic keypoint matching aims at establishing correspondence between keypoint sets in images. As graphs are powerful tools to represent points and their complex relationships, graph matching provides an effective way to find desired semantic keypoint correspondences. Recent deep graph matching methods have shown excellent performance, but there is still a lack of exploration and utilization of spatial information of keypoints as nodes in graphs. More specifically, existing methods are insufficient to capture the relative spatial relations through current graph construction approaches from the locations of semantic keypoints. To address these issues, we introduce a positional reconstruction encoder-decoder (PR-EnDec) to model intrinsic graph spatial structure, and present an end-to-end graph matching network PREGM based on PR-EnDec. Our PR-EnDec consists of a positional encoder that learns effective node spatial embedding with the affine transformation invariance, and a spatial relation decoder that further utilizes the high-order spatial information by reconstructing the locational structure of graphs contained in the node coordinates. Extensive experimental results on four public keypoint matching datasets demonstrate the effectiveness of our proposed PREGM.

## 1   Introduction

Image matching is a common and basic matching problem in multimedia and visual applications. The goal of image matching is to find a dense semantic content correspondence between two images, while feature-based matching is a typical strategy to solve the above problems, and in feature matching,

---

[*]Correspondence: `lvxiaoqing@pku.edu.cn`. Author emails: `{chnzyx,jiaruiqi}@pku.edu.cn`, `linhongxiang@stu.pku.edu.cn`, `920106840131@njust.edu.cn`, `zhaoyumeng_311@bupt.edu.cn`, `lvxiaoqing@pku.edu.cn`

37th Conference on Neural Information Processing Systems (NeurIPS 2023).

the semantic keypoints in images are the most commonly used matching targets. Semantic keypoint matching aims at establishing correspondence between keypoint sets in images, and it has been applied to a wide range of applications, such as object tracking [35, 24], image retrieval [40, 11], and pose estimation [39, 3], etc.

The semantic keypoint data has the properties of non-Euclidean and unstructured for its disorder and discreteness, and graphs are powerful tools to represent some complex objects and their interactions. Specifically, we take each semantic keypoint as a node, and build edges by heuristic methods according to the positions of nodes to construct a corresponding graph. Graph matching is an effective method to solve the correspondence between the semantic keypoints in two images.

As deep learning models are scalable and time-efficient, Graph Neural Networks (GNNs) are increasingly utilized in graph matching tasks, such as GMN [44], PCA [32], and BBGM [29], etc.

However, most graph matching approaches rely on the matching of basic items of graphs, i.e. computing affinities of nodes and edges, lack the mining and utilization of spatial context information hidden in the locations of keypoints as nodes of graphs. Recently, in task scenarios such as point cloud and knowledge graph, which the nodes in graphs have locational information, there are methods such as [21, 18] utilizing positional encoder to add the spatial information as additional features. However, current methods in graph matching domain lack such spatial positional information supplementation. As the positional information is closer to the intrinsic characteristic of the graph, it is important to integrate spatial descriptions into graph feature embeddings in graph matching.

But utilizing underlying spatial information faces the following challenges: 1) Through taking key points as graph nodes and edges are associated with pairwise distance, such as delaunay triangulation, it is not sufficient to mine hidden relative positional relationship. 2) The description of spatial information from coordinates of keypoints is relatively simple compared with conventional visual features, so it needs more refined network structure to learn location information, in order to facilitate subsequent feature fusion.

To address the two challenges mentioned above, we propose a positional reconstruction encoder-decoder network (PR-EnDec) in our PREGM by integrating more spatial information into semantic features and learning more distinguishable node features. The positional encoder in PR-EnDec aims to learn the encoding of node positional information with spatial invariance, specifically, the affine transformation invariance of node coordinates. The spatial relation decoder in PR-EnDec is designed to recover the locational structure from the learned features of the encoder, and it reconstructs the relative distances and areas among the sampled nodes. To implement the joint learning of PR-EnDec about the intrinsic positional features in graphs, we, consequently, propose a corresponding contrastive loss function and a reconstruction loss function in the encoder and the decoder respectively.

Our main contributions are summarized as follows:

- We propose an end-to-end graph matching network, PREGM, which learns more distinguishable node features, not only captures visual information in images but also models intrinsic spatial structure of keypoints by our designed PR-EnDec.

- Our PR-EnDec consists of a positional encoder that learns effective graph positional encoding with affine transformation invariance, and a spatial relation decoder used to enhance such learning by reconstructing the graph locational information such as relative distances and areas from learned positional encoding. We also design a corresponding contrastive loss function and a reconstruction loss function in the encoder and decoder respectively to help learn effective node positional characteristics.

- We evaluate our PREGM on four public keypoint matching datasets. Our experimental results demonstrate that PREGM outperforms state-of-the-art methods. We also present an ablation study, which shows the effectiveness of each component of PREGM.

# 2 Related Work

## 2.1 Graph Matching

Graph matching is always an effective strategy to find correspondence between semantic keypoints as graphs have sufficient representation ability for points and their complex relationships, and it can then be applied to downstream tasks such as image matching and understanding. As it is famous for being one of the practically most difficult NP-complete problems, researchers often use optimization algorithms to find an acceptable suboptimal solution for the graph matching problem. Traditional graph matching methods [17, 5, 7, 20, 36] generally use heuristic search algorithms to find suboptimal solutions.

Recent graph matching methods begin to combine combinatorial optimization with deep learning models. GMN [44] proposes an end-to-end graph matching framework, calculates the unary and pairwise affinities of nodes through CNN features, and also uses spectral matching as a differentiable solver to combine deep learning with graph matching. PCA [32] uses graph neural network (GNN) to learn and aggregate graph structure information into node feature representation, and introduces Sinkhorn network [1] as the combinatorial solver for finding node correspondence. BBGM [29] presents an end-to-end trainable architecture that incorporates a state-of-the-art differentiable combinatorial graph matching solver, and introduces a global attention mechanism to improve the matching performance.

Although the research on graph matching has made the above progress, the existing methods lack the distinguishing and informative graph spatial feature descriptions, and adaptive adjustment of node correspondence on graph matching. In this paper, we overcome these limitations by proposing a positional reconstruction encoder-decoder (PR-EnDec) and designing a contrastive learning and positional feature reconstruction procedure to guide the learning of graph matching.

## 2.2 Positional Encoding on Graphs

Positional Encoding was first proposed in the Transformer [31] model. In general, positional encoding refers to the process of encoding spatial relationships or topological information of elements as supplementary information in order to augment the representational capacity of a model. The categorization of positional encoding can be further elaborated as follows: spatial positional encoding utilizes point location as input, while topological positional encoding employs the graph structure as input.

For spatial positional encoding, the point location is used as input to obtain a high-dimensional embedding that contains effective locational information. Space2Vec [21] introduces *theory* and *grid* as a 2D multi-scale spatial positional encoder by using sinusoidal functions with different frequencies. PointCNN [18] designs a point set spatial positional encoder with Point Conv layers consisting of sampling layer, grouping layer, and aggregation layer. DGCNN [38] considers both global shape structure and local neighborhood information, and proposes spatial positional encoder layers containing dynamic graph neighborhood and edge convolution module. This paper proposes a spatial position encoding on graphs by leveraging the characteristic of node representation containing coordinate information in image keypoint matching.

Topological positional encoding is specific to graph matching, where the graph structure is used as input to obtain the topological embedding of the nodes. Maskey et al. [22] generalizes Laplacian-based positional encoding by defining the Laplace embedding to more general dissimilarity functions such as p-norm rather than the 2-norm used in the original formulation. INFMCS [16] designs a permutation-invariant node ordering based on closeness centrality, and effectively enhances the representation of graph structural features.

In recent years, there has also been research on spatial positional encoding in the field of graph matching. GCAN [13] utilizes SplineCNN [10] to encode positional information in 2D spatial space, and successfully capture structural information of the graph.

However, there remains the open question of how to further mine intrinsic spatial node information in graphs. In this paper, we propose a positional reconstruction encoder-decoder architecture to obtain effective node spatial positional encoding using 2D point location.

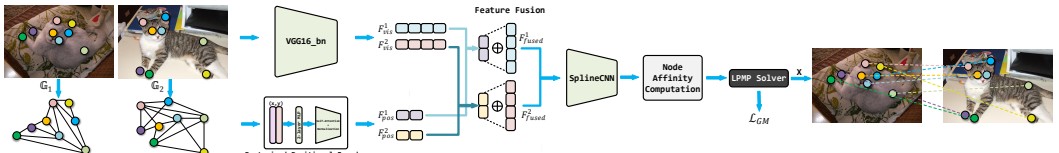

Figure 1: **The framework of our PREGM.** The visual features are extracted by vgg16_bn from images, and the positional encoding is obtained by our pretrained positional encoder from graphs. The visual features and positional encoding of nodes are further fused to be fed into the following modules. Next, the fused features are updated and refined by the message passing module SplineCNN. Finally, we compute node affinity from updated features and adopt a differentiable graph matching solver LPMP to find the correspondence matrix X.

## 3 Problem Formulation of Graph Matching

Given the keypoints in the images, each node is generally associated with a keypoint, and edges are built according to the spatial position relationship between nodes when building an undirected graph. The graph is represented by $\mathbb{G} = (\mathbb{V}, \mathbb{E}, \mathcal{V}, \mathcal{E})$:

- $\mathbb{V} = \{v_1, ..., v_n\}$ denotes the node set.
- $\mathbb{E} \subseteq \mathbb{V} \times \mathbb{V}$ denotes the edge.
- $\mathcal{V} = \{\mathbf{v_i} | \mathbf{v_i} \in \mathbb{R}^{d_v}, i = 1, 2, ..., |\mathbb{V}|\}$ denotes the node feature set.
- $\mathcal{E} = \{\mathbf{e_i} | \mathbf{e_i} \in \mathbb{R}^{d_e}, i = 1, 2, ..., |\mathbb{E}|\}$ denotes the edge feature set.

We use the adjacency matrix $\mathbb{A}$ to represent the connections of nodes in an undirected graph $\mathbb{G}$, that is, $\mathbb{A}_{ij} = 1$ iff there is an edge $e_{ij} = (v_i, v_j) \in \mathbb{E}$.

Given two graphs $\mathbb{G}_1 = (\mathbb{V}_1, \mathbb{E}_1, \mathcal{V}_1, \mathcal{E}_1)$ and $\mathbb{G}_2 = (\mathbb{V}_2, \mathbb{E}_2, \mathcal{V}_2, \mathcal{E}_2)$, where $|\mathbb{V}_1| = |\mathbb{V}_2| = n$, the goal of graph matching is to find the correspondence matrix $\mathbf{X} \in \{0, 1\}^{n \times n}$ between nodes $\mathbb{V}_1$ and $\mathbb{V}_2$ of the two graphs $\mathbb{G}_1$ and $\mathbb{G}_2$, wherein $\mathbf{X}_{ij} = 1$ iff $v_i \in \mathbb{V}_1$ corresponds to $v_j \in \mathbb{V}_2$.

The ultimate goal of the graph matching problem is to find the optimal node correspondence between two graphs $\mathbb{G}_1$ and $\mathbb{G}_1$, so the graph matching problem is often considered as a quadratic assignment programming problem (QAP problem):

$$\mathbf{x}^* = \text{argmax}_{\mathbf{x}} \mathbf{x}^T \mathbf{K} \mathbf{x}, \tag{1}$$

where $\mathbf{x} = vec(\mathbf{X}) \in \{0, 1\}^{n^2}$, $\mathbf{X}\mathbf{1}_n = \mathbf{1}_n$, and $\mathbf{X}^T\mathbf{1}_n = \mathbf{1}_n$. $\mathbf{x}^*$ denotes the desired node correspondence, and $\mathbf{1}_n$ denotes a vector of $n$ ones. $\mathbf{K} \in \mathbb{R}^{n^2 \times n^2}$ is the corresponding affinity matrix:

$$\mathbf{K}_{ia,jb} = \begin{cases} s_{ia}^v, & \text{if } i = j \,\& \, a = b, \\ s_{ia,jb}^e, & \text{else if } \mathbb{A}_{ij}^1 \mathbb{A}_{ab}^2 > 0, \\ 0, & \text{otherwise.} \end{cases} \tag{2}$$

where $s_{ia}^v$ represents the affinity between node features $\mathbf{v}_i \in \mathcal{V}_1$ and $\mathbf{v}_a \in \mathcal{V}_2$, $s_{ia,jb}^e$ represents the affinity between edge features $\mathbf{e}_{ij} \in \mathcal{E}_1$ and $\mathbf{e}_{ab} \in \mathcal{E}_2$.

## 4 Methodology

The framework of our PREGM Network is illustrated in Figure 1. The positional encoder module of pretrained PR-EnDec takes the coordinates of keypoints as input. It extracts a positional encoding vector for each keypoint in the images. Next, the positional encoding and the visual features extracted by vgg16_bn are further fused. Then, the fused features are fed into the message-passing module to obtain the refined node feature. Finally, the node affinity is computed and processed by the LPMP solver, generating the node correspondence matrix X introduced in the problem formulation.

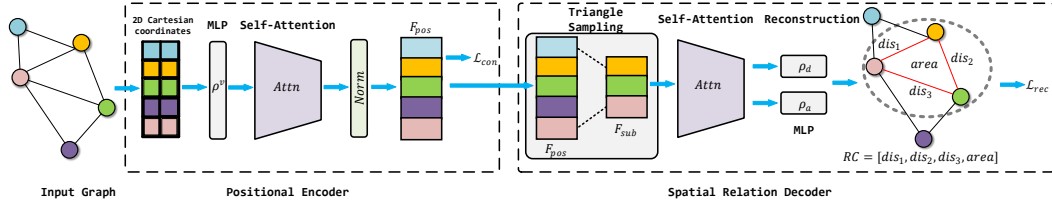

Figure 2: The framework of our PR-EnDec network and the detailed structure of its positional encoder and spatial relation decoder.

As an independent stage, we pretrain the abovementioned positional encoder along with the spatial relation decoder module of PR-EnDec (as shown in Figure 2). The pretraining stage enables the positional encoder to learn the high-order spatial information. The PR-EnDec consisting two modules,

**Positional Encoder.** This module takes the coordinates of keypoints as input and learns positional encoding. We first obtain the high-dimensional coordinate embedding vectors by an MLP. The vectors are then fed into self-attention blocks and a normalization layer to learn positional encoding. The attention mechanism provides sequence independence and relative information of nodes. Besides, we propose a contrastive loss for the encoder module. The loss function ensures the affine transformation invariance and learns the relative position information.

**Spatial Relation Decoder.** To reconstruct the spatial structure of graphs, this module generates triangle assignments by distance-based random sampling. In each triangle assignment, the module takes positional encoding corresponding to the three keypoints as the input of self-attention blocks. Next, the processed features are fed into MLPs, and the relative spatial relations of the three points are reconstructed. We adopt a Mean Squared Error (MSE) loss to compare the ground truth relative spatial relations and the decoder's predictions.

### 4.1 Positional Encoder

The Positional Encoder only takes geometric features of keypoints as inputs,

$\mathcal{V}^g = \{v_i^g | v_i^g \in \mathbb{R}^{d_v^g}, i = 1, ..., n\}$ denotes the node geometric feature set.

Specifically, we utilize the 2D Cartesian coordinates of each node $v_i$ as its geometric feature $v_i^g = [x_i, y_i]$. The coordinate embedding sub-module embeds the geometric feature $\mathcal{V}^g$ into latent representations by a two-layer MLP, denoted as $\rho^v$. The updated geometric graph $\mathbb{G}^g = (\mathbb{V}, \rho^v(\mathcal{V}^g))$ is then fed into the following attention sub-module.

The attention sub-module consists of $l_m$ attention blocks (denoted as $Attn_i^e$) and a normalization layer (denoted as $Norm$), which updates latent representations of $\mathbb{G}^g$. An attention block is composed of a multi-head self-attention layer and a feed-forward layer, which captures the spatial feature of $\mathbb{G}^g$ by aggregating the latent representations of each node.

$$\mathcal{V}_0^g = \rho^v(\mathcal{V}^g), \tag{3}$$

$$\mathcal{V}_i^g = Attn_i^e(\mathcal{V}_{i-1}^g), i = 1, 2, ..., l_m. \tag{4}$$

The self-attention mechanism is sequence-independent, which is well suited for extracting high-order relative position information between nodes. There is a normalization layer after attention blocks to allow the positional encoding of nodes to be better fused with visual features nodes on the graph matching task. The final positional encoding obtained by the encoder module is denoted as $F_{pos}$.

$$F_{pos} = Norm(\mathcal{V}_{l_m}^g). \tag{5}$$

**Encoder Loss.** For two given graphs $\mathbb{G}_1 = (\mathbb{V}_1, \mathbb{E}_1, \mathcal{V}_1, \mathcal{E}_1)$ and $\mathbb{G}_1 = (\mathbb{V}_1, \mathbb{E}_1, \mathcal{V}_1, \mathcal{E}_1)$, the positional encoder generates the positional encoding denoted as $F_{pos}^1 = \{f_i^1 | i = 1, 2, ..., |\mathbb{V}|\}$ and $F_{pos}^2 = \{f_i^2 | i = 1, 2, ..., |\mathbb{V}|\}$ respectively. We expect that for all matched node $i$ in graph $\mathbb{G}_1$ and node $j$ in graph $\mathbb{G}_2$, the corresponding node positional encoding $f_i^1$ and $f_j^2$ should be similar. To let the encoder learn the relative position information of the nodes, we perform contrastive learning that

perseves the affine invariance of the positional encoding. Specifically, we adopt negative samples to conduct contrastive learning to avoid the encoder module outputs trivial node features that are all similar.

Therefore, we classify $(\mathbb{G}_1, \mathbb{G}_2)$ as a positive graph pair if $\mathcal{V}_2^g$ is affine transformed by $\mathcal{V}_1^g$ or the keypoints in $\mathcal{V}_1^g$ and $\mathcal{V}_2^g$ are one-by-one matched. On the contrary, $(\mathbb{G}_1, \mathbb{G}_2)$ is a negative graph pair if the keypoints are not matched.

For each training batch, the positive graph pair set $\mathbb{GS}_p$ and negative graph pair set $\mathbb{GS}_n$ are generated by node permuting or affine transformation. Let $\mathbb{FS}_p$ and $\mathbb{FS}_n$ denote the corresponding positional encoding pair set of $\mathbb{GS}_p$ and $\mathbb{GS}_n$ respectively, we propose a contrastive loss $\mathcal{L}_{con}$,

$$S_p = \sum_{(F_{pos}^1, F_{pos}^2) \in \mathbb{FS}_p} exp(\tau * sim(F_{pos}^1, F_{pos}^2)), \tag{6}$$

$$S_n = \sum_{(F_{pos}^1, F_{pos}^2) \in \mathbb{FS}_n} exp(\tau * sim(F_{pos}^1, F_{pos}^2)), \tag{7}$$

$$\mathcal{L}_{con} = -Log(\frac{S_p}{S_p + S_n}), \tag{8}$$

where $S_p$ and $S_n$ denote the sum of the exponential of positive/negative pairs' similarity, respectively. Additionally, $\tau$ denotes the temperature constant, and $sim()$ denotes the cosine similarity.

## 4.2 Spatial Relation Decoder

To learn the high-order spatial information, we propose $k$-th-order geometric reconstruction assignments. Sampling $k$ points from the node set as an assignment, the corresponding geometric feature set is denoted as $\mathcal{V}_s^g$ and positional encoding set as $F_{sub} \subseteq F_{pos}$. If a decoder $Dec$ takes $F_{sub}$ as input, we claim the decoder provides $k$-th-order relative position information if there exists a non-degenerate geometric function $\mathcal{F}$, $\mathcal{F}(\mathcal{V}_s^g) \approx Dec(F_{sub})$. Specifically, we choose $k = 3$ and function $\mathcal{F}$ with Euclidean distance and triangle area to learn the third-order relative position information.

The decoder module first samples three keypoints and obtains the corresponding geometric feature set $\mathcal{V}_s^g = \{v_a^g, v_b^g, v_c^g\}$ and positional encoding set $F_{sub} = \{f_a, f_b, f_c\}$. The random sampling method is related to the Euclidean distance from the previous sampled node to learn the relative information better, for the sampling method guarantees that the distances between the sampled nodes are not too far. Next, the sampled feature set $F_{sub}$ is fed into $l_n$ attention blocks. The intermediate feature is denoted as $\hat{F}_{sub} = \{\hat{f}_a, \hat{f}_b, \hat{f}_c\}$. We utilize 2 MLPs, denoted as $\rho_d$ and $\rho_a$, to approximate Euclidean distance and triangle area function, respectively. The reconstructed relative position information of the decoder is represented as:

$$RC = [\rho_d(\hat{f}_a, \hat{f}_b), \rho_d(\hat{f}_b, \hat{f}_c), \rho_d(\hat{f}_a, \hat{f}_c), \rho_a(\hat{f}_a, \hat{f}_b, \hat{f}_c)]. \tag{9}$$

Let $Dis$ denotes Euclidean distance and $Area$ denotes triangle area function, $RC$'s corresponding geometric function $\mathcal{F}$ is obviously

$$\mathcal{F}(\mathcal{V}_s^g) = [dis1, dis2, dis3, area], \tag{10}$$

where $dis1 = Dis(v_a^g, v_b^g), dis2 = Dis(v_b^g, v_c^g), dis3 = Dis(v_a^g, v_c^g), area = Area(v_a^g, v_b^g, v_c^g)$ denote ground-truth third-order relative position information for the decoder to reconstruct.

**Decoder Loss.** Since we calculate the approximated relative position information $RC$ and the ground-truth $RC^{gt} = \mathcal{F}(\mathcal{V}_s^g)$, we propose the reconstruction loss $\mathcal{L}_{rec}$:

$$\mathcal{L}_{rec} = \text{MSE}(RC, RC^{gt}). \tag{11}$$

**PR-EnDec Loss.** Finally, we combine the two losses to guide the training of our PR-EnDec jointly,

$$\mathcal{L}_{PR-EnDec} = \mathcal{L}_{con} + \lambda \cdot \mathcal{L}_{rec}, \tag{12}$$

where $\lambda$ controls the relative importance of $\mathcal{L}_{rec}$.

Through pretraining of PR-EnDec, we obtain an effective encoder for extracting spatial positional encoding for utilization in subsequent PREGM.

### 4.3 Graph Matching with PR-EnDec

Our training procedure is divided into two stages: In the first stage, we pretrain our PR-EnDec, and in the second stage of graph matching, our positional encoder module of PR-EnDec serves as a sub-module of the base graph matching model, providing learned positional encoding in the first stage.

After the training of PR-EnDec, we freeze all parameters of the encoder module and put it in the training of graph matching tasks. The positional encoding $F_{pos}$ generated by the positional encoder module is fused with visual features $F_{vis}$ extracted by a standard CNN:

$$F_{fused} = Linear(F_{pos}) + F_{vis}. \tag{13}$$

where $Linear$ denotes a linear layer.

Next, we feed the fused features into a message-passing module to produce the refined node features, and the message-passing module is implemented by a two-layer SplineCNN [10]. After the message-passing module, node and edge affinities are computed and passed to the differentiable graph matching solver LPMP in [25]. The resulting correspondence matrix X is compared to the ground truth $X^{gt}$ and the loss function is their Hamming distance:

$$\mathcal{L}_{GM} = X \cdot (1 - X^{gt}) + X^{gt} \cdot (1 - X). \tag{14}$$

So far, the graph matching problem formulated previously is solved by our PREGM with the two sequential training phases: the PR-EnDec pretrain and the graph matching task.

## 5 Experiments

We conduct experiments on four public keypoint matching datasets and verify the effectiveness of each component of our PREGM by ablation study. We introduce the datasets, baselines, implementation details, and then report the results. The results demonstrate that our PREGM consistently outperforms all other approaches.

### 5.1 Datasets

We evaluate our PREGM on four public keypoint matching datasets: PascalVOC [8], Willow ObjectClass [4], SPair-71k [23], and IMC-PT-SparseGM [14].

The PascalVOC dataset includes 20 classes of keypoints with Berkeley annotations [2] and images with bounding boxes. The PascalVOC dataset is relatively challenging, since the scale, pose and illumination of instances in images are rich and diverse, and the number of annotated keypoints in each image also varies from 6 to 23. When conducting experiments on the PascalVOC dataset, we follow the standard protocol [32, 37]: First, each object is cropped according to its corresponding bounding box and scaled to 256 × 256 px. Second, we use 7,020 images for training and 1,682 for testing.

The Willow ObjectClass dataset contains images of five categories: face, duck, winebottle, car, and motorbike, the first three categories are from the Caltech-256 dataset [12], and the last two categories are from the PascalVOC 2007 dataset [8]. Each category contains at least 40 different images, and each image is labeled with 10 distinctive keypoints on the target object to be matched. Following the default setting in [4, 32], we crop the images to the bounding boxes of the objects and rescale to 256 × 256 px, 20 images of each class are selected during training, and the rest are for testing.

The SPair-71k dataset is a relatively novel dataset, which was recently published in the paper [23] about dense image matching. Spair-71k contains 70958 image pairs of 18 categories from PascalVOC 2012 dataset [8] and Pascal 3D+ dataset [41]. Compared with the other two datasets, SPair71-k has the advantages of higher image quality and richer annotations which includes detailed semantic keypoints, object bounding boxes, viewpoints, scales, truncation, and occlusion differences of image pairs. In addition, compared with PascalVOC dataset, SPair-71k removes two classes with ambiguous and poor annotations: sofa and dining table. Following [29], we use 53,340 image pairs for training, 5,384 for validation, and 12,234 for testing, and we also scale each image to 256× 256 px.

The IMC-PT-SparseGM dataset contains 16 object categories and 25061 images [14], which gather from 16 tourist attractions around the world. The IMC-PT-SparseGM benchmark involves matching

Table 1: Matching accuracy (%) on PascalVOC dataset.

| Method | aero | bike | bird | boat | bottle | bus | car | cat | chair | cow | dtable | dog | horse | mbike | person | plant | sheep | sofa | train | tv | Avg |
|---|---|---|---|---|---|---|---|---|---|---|---|---|---|---|---|---|---|---|---|---|---|
| GNCCP | 28.9 | 37.1 | 46.2 | 53.1 | 48.0 | 36.3 | 45.5 | 34.7 | 36.3 | 34.2 | 25.2 | 35.3 | 39.8 | 39.6 | 40.7 | 61.9 | 37.4 | 50.5 | 67.0 | 34.8 | 41.6 |
| ABPF | 30.9 | 40.4 | 47.3 | 54.5 | 50.8 | 35.1 | 46.7 | 36.3 | 40.9 | 38.9 | 16.3 | 34.8 | 39.8 | 39.6 | 39.3 | 63.2 | 37.9 | 50.2 | 70.5 | 41.3 | 42.7 |
| GMN | 31.9 | 47.2 | 51.9 | 40.8 | 68.7 | 72.2 | 53.6 | 52.8 | 34.6 | 48.6 | 72.3 | 47.7 | 54.8 | 51.0 | 38.6 | 75.1 | 49.5 | 45.0 | 83.0 | 86.3 | 55.3 |
| PCA | 40.9 | 55.0 | 65.8 | 47.9 | 76.9 | 77.9 | 63.5 | 67.4 | 33.7 | 65.5 | 63.6 | 61.3 | 68.9 | 62.8 | 44.9 | 77.5 | 67.4 | 57.5 | 86.7 | 90.9 | 63.8 |
| CIE | 51.2 | 69.2 | 70.1 | 55.0 | 82.8 | 72.8 | 69.0 | 74.2 | 39.6 | 68.8 | 71.8 | 70.0 | 71.8 | 66.8 | 44.8 | 85.2 | 69.9 | 65.4 | 85.2 | 92.4 | 68.9 |
| NGM | 50.8 | 64.5 | 59.5 | 57.6 | 79.4 | 76.9 | 74.4 | 69.9 | 41.5 | 62.3 | 68.5 | 62.2 | 62.4 | 64.7 | 47.8 | 78.7 | 66.0 | 63.3 | 81.4 | 89.6 | 66.1 |
| DGMC | 50.4 | 67.6 | 70.7 | 70.5 | 87.2 | 85.2 | 82.5 | 74.3 | 46.2 | 69.4 | 69.9 | 73.9 | 73.8 | 65.4 | 51.6 | 98.0 | 73.2 | 69.6 | 94.3 | 89.6 | 73.2 |
| EAGM | 49.4 | 62.1 | 64.6 | 75.3 | 90.9 | 80.9 | 71.1 | 61.3 | 48.7 | 65.9 | 87.5 | 58.4 | 66.3 | 60.1 | 56.3 | 97.1 | 64.7 | 60.6 | 96.0 | 93.0 | 70.5 |
| BBGM | 61.5 | 75.0 | 78.1 | 80.0 | 87.4 | 93.0 | 89.1 | 80.2 | 58.1 | 77.6 | 76.5 | 79.3 | 78.6 | 78.8 | 66.7 | 97.4 | 76.4 | 77.5 | 97.7 | 94.4 | 80.1 |
| DIP-GM | 58.5 | 74.9 | 76.2 | 76.0 | 87.1 | **94.7** | 89.8 | 79.8 | 60.4 | 77.5 | 79.8 | 78.0 | 76.2 | 78.2 | 64.3 | 97.1 | 76.4 | 76.4 | 96.5 | 93.2 | 79.6 |
| IA-NGM-v2 | 61.5 | 73.8 | 74.0 | 79.4 | 89.1 | 94.6 | 89.7 | 77.5 | 67.1 | 77.3 | 92.4 | 76.8 | 77.1 | 77.4 | 65.8 | 98.5 | 77.5 | 79.5 | 96.6 | 92.3 | 80.9 |
| ASAR | 62.9 | 74.3 | 79.5 | **80.1** | 89.2 | 94.0 | 88.9 | 78.9 | 58.8 | 79.8 | 88.2 | 78.9 | 79.5 | 77.9 | 64.9 | 98.2 | 77.5 | 77.1 | 98.6 | 93.7 | 81.1 |
| COMMON | **65.6** | 75.2 | **80.8** | 79.5 | 89.3 | 92.3 | 90.1 | 81.8 | 61.6 | 80.7 | **95.0** | 82.0 | 81.6 | 79.5 | 66.6 | **98.9** | **78.9** | 80.9 | **99.3** | 93.8 | 82.7 |
| Ours | 62.6 | **76.4** | 79.3 | 78.7 | 90.3 | 94.1 | **92.3** | **83.4** | **71.9** | **82.3** | 92.4 | **82.4** | **82.0** | **85.0** | **69.2** | 98.7 | 78.4 | **90.7** | 96.7 | **95.1** | **84.1** |

Table 2: Matching accuracy (%) on Willow dataset.

| Method | PT | WT | car | duck | face | mbike | wbottle | Avg |
|---|---|---|---|---|---|---|---|---|
| GNCCP | × | × | 86.4 | 77.4 | **100.0** | 95.6 | 95.7 | 91.0 |
| ABPF | × | × | 88.4 | 80.1 | **100.0** | 96.2 | 96.7 | 92.3 |
| GMN | × | × | 74.3 | 82.8 | 99.3 | 71.4 | 76.7 | 80.9 |
| PCA | × | × | 84.0 | 93.5 | **100.0** | 76.7 | 96.9 | 90.2 |
| CIE | × | × | 82.2 | 81.2 | **100.0** | 90.0 | 97.6 | 90.2 |
| NGM | × | ✓ | 84.1 | 77.4 | 99.2 | 82.1 | 93.5 | 87.3 |
| DGMC | × | ✓ | **98.3** | 90.2 | **100.0** | 98.5 | 98.1 | 97.0 |
| DGMC | ✓ | ✓ | 96.5 | 93.2 | **100.0** | 98.8 | 99.9 | 97.7 |
| EAGM | × | ✓ | 94.4 | 89.7 | **100.0** | 99.3 | 99.2 | 96.5 |
| BBGM | × | ✓ | 96.9 | 89.0 | **100.0** | 99.2 | 98.8 | 96.8 |
| BBGM | ✓ | ✓ | 95.7 | 93.1 | **100.0** | 98.9 | 99.1 | 97.4 |
| Ours | × | ✓ | 97.4 | 96.8 | **100.0** | **99.8** | 96.8 | 98.2 |
| Ours | ✓ | ✓ | 97.8 | **99.0** | **100.0** | 99.6 | **100.0** | **99.3** |

larger scenes and take a step closer to the real-world downstream tasks. We take 13 classes as the training set and the other 3 classes as the test set. Experiments are conducted on the benchmark with 50 anchors.

## 5.2 Baselines and Metrics

We compare our PREGM with the state-of-the-art methods, including two types of baselines: (1) 2 non-learning methods GNCCP [20], ABPF [36]; (2) 11 learning-based methods GMN [44], PCA [32], CIE [43], NGM [33], DGMC [9], EAGM [27], BBGM [29], DIP-GM [42], IA-NGM-v2 [26], ASAR [28], COMMON [19]. For a fair comparison, we follow previous work [32] and extract node visual initial features from relu4_2 and relu5_1 of vgg16_bn [30] via feature alignment. For the input image pairs, we construct the edges of graphs by Delaunay triangulation [6]. We adopt a common metric matching accuracy to evaluate the experimental results, which is computed as the number of correctly matched keypoint pairs averaged by the total number of all true matched keypoint pairs.

## 5.3 Experimental Settings

In our implementation, we build up our PREGM based on the state-of-the-art method [29], which presents an end-to-end trainable architecture that incorporates a state-of-the-art differentiable combinatorial graph matching solver. Our codes will be available in the future. In all experiments, we use the same set of hyperparameters. We employ Adam [15] optimizer with an initial learning rate of $1 \times 10^{-4}$ for PR-EnDec, and $9 \times 10^{-4}$ for other models, and the learning rate is halved every three epochs. We empirically set $l_m = 3$ and $l_n = 2$ in the encoder and decoder, and choose temperature constant $\tau = 2$ in $L_{con}$, and balance factor $\lambda = 1/32$ in $L_{PR-EnDec}$. We also set batch size = 8 for PascalVOC, Willow ObjectClass, and Spair-71k datasets. Our learning procedure is divided into two stages: The pre-training of the PR-EnDec is uniformly conducted on the PascalVOC dataset. Then the graph matching framework is trained on each dataset, while the parameters of the positional encoder are fixed. All experiments are run on a single GTX-1080Ti GPU.

Table 3: Matching accuracy (%) on Spair-71k dataset.

| Method | aero | bike | bird | boat | bottle | bus | car | cat | chair | cow | dog | horse | mbike | person | plant | sheep | train | tv | Avg |
|---|---|---|---|---|---|---|---|---|---|---|---|---|---|---|---|---|---|---|---|
| GMN | 50.1 | 40.4 | 62.7 | 46.8 | 60.8 | 66.8 | 57.2 | 62.0 | 40.5 | 61.5 | 49.5 | 46.5 | 56.5 | 43.9 | 76.1 | 44.7 | 61.2 | 75.7 | 55.7 |
| PCA | 58.9 | 42.3 | 72.1 | 54.1 | 61.2 | 77.3 | 66.1 | 65.2 | 50.4 | 64.9 | 56.8 | 55.5 | 64.3 | 53.4 | 86.2 | 49.1 | 75.5 | 91.4 | 63.6 |
| DGMC | 54.8 | 44.8 | 80.3 | 70.9 | 65.5 | 90.1 | 78.5 | 66.7 | 66.4 | 73.2 | 66.2 | 66.5 | 65.7 | 59.1 | 98.7 | 68.5 | 84.9 | 98.0 | 72.2 |
| BBGM | 66.9 | 57.7 | 85.8 | 78.5 | 66.9 | 95.4 | 86.1 | 74.6 | 68.3 | 78.9 | 73.0 | 67.5 | 79.3 | 73.0 | 99.1 | 74.8 | 95.0 | 98.6 | 78.9 |
| DIP-GM | 63.7 | 54.5 | 89.0 | 80.9 | 64.2 | 95.0 | 87.3 | 73.5 | 71.0 | 79.7 | 73.4 | 68.1 | 75.1 | 71.2 | 98.8 | 76.9 | 96.0 | 99.2 | 78.7 |
| Ours | **71.3** | **61.0** | **89.6** | **82.0** | **68.4** | **98.4** | **91.5** | **75.1** | **77.6** | **84.1** | **77.3** | **74.9** | **83.4** | **74.8** | **99.5** | **77.6** | **97.5** | **99.8** | **82.4** |

Table 4: Matching accuracy (%) on the IMC-PT-SparseGM.

| Method | Reichstag | Sacre_coeur | St_peters_square | Avg |
|---|---|---|---|---|
| GANN-GM | 76.0 | 44.2 | 50.5 | 56.9 |
| BBGM | 99.1 | 79.5 | 86.8 | 88.4 |
| Ours | **99.8** | **84.8** | **87.6** | **90.7** |

## 5.4 Performance Evaluation

We conduct experiments on four public datasets: PascalVOC, WillowObject Class, Spair-71k and IMC-PT-SparseGM for the keypoint matching problem, and follow the most common experimental setting, where intersection filtering is applied to generate graphs with the equal size.

Firstly, we report the matching accuracy of the 20 classes and average accuracy on PascalVOC dataset in Table 1, where the best results are shown in bold. The results demonstrate that our model PREGM performs much better than all traditional graph matching methods, and also achieves better performance against state-of-the-art deep graph matching models with matching accuracy 84.1%. And in the 20 classes of PasalVOC dataset, our PREGM has achieved the best results in the other 17 classes except for the boat, bottle and train class. As there are large differences in the scale, pose and illumination of matching objects, PascalVOC is a complex matching dataset, thus the matching accuracy of traditional graph matching methods is not more than 50%, and the performance of deep graph matching methods is relatively better. Our PREGM learns effective node spatial characteristics by PR-EnDec, so as to further improve the graph matching performance.

For the relatively simple WillowObject Class dataset, as shown in Table 2, there are two different training strategies: PT and WT, which PT means matching frameworks are pre-trained on PascalVOC, and WT means learning models are then fine-tuned on WillowObject Class Dataset. In this paper, we adopt two strategies: PT only, both PT and WT, and in both situations, our model achieves the best performance among other state-of-the-art methods with matching accuracy 98.2% and 99.3%. The results again demonstrate the effectiveness of learning positional node features hidden in graphs in our PREGM.

We also conduct experiments on Spair-71k dataset, as shown in Table 3, we compare our PREGM with deep learning methods GMN, PCA, DGMC, BBGM, and DIP-GM. Our PREGM performs best with matching accuracy 81.9%, and shows superiority in total 18 classes, which demonstrates the generalization ability of our model on different objects. SPair-71k dataset, as a relatively new dataset, has the advantages of high image quality, rich annotation, and fixed image pairs for matching, which is a more convincing dataset. Thus our PREGM achieves the best results on the SPair-71k dataset, which further proves the effectiveness of our PREGM.

Additionally, we evaluate our model on the IMC-PT-SparseGM dataset, as shown in Table 4, our model demonstrates outstanding performance on this demanding benchmark. The results outshine the performance of the other methods, GANN-GM [34] and BBGM, by a significant margin. In terms of the average accuracy across these landmarks, our model excels with an impressive mean accuracy of 90.7%. The IMC-PT-SparseGM dataset is characterized by its substantial number of images, nodes, and high partial rate, making it one of the most comprehensive benchmarks for visual graph matching. Our approach, showcased in these results, demonstrates its ability to handle larger scenes and move closer to real-world applications, such as structure from motion.

Table 5: Ablation Study on PascalVOC dataset.

| Method | baseline | w/o positional encoder | w/o spatial relation decoder | w/o recon_area | w/o recon_dis | w/o visual features |
|---|---|---|---|---|---|---|
| Accuracy | **84.1%** | 83.4% | 81.7% | 83.8% | 82.7% | 76.8% |

Table 6: Parameter analysis of learning rate on PascalVOC.

| lr ($\times 10^{-4}$) | 6 | 7 | 8 | 9 | 10 | 11 | 12 |
|---|---|---|---|---|---|---|---|
| Accuracy | 83.2% | 83.7% | 83.8% | **84.1%** | 83.9% | 83.7% | 83.7% |

## 5.5 Ablation Study and Parameter Analysis

To evaluate the effect of each component in our framework, we conduct comprehensive ablation studies with/without positional encoder, spatial relation decoder, and we also consider the two separate cases of no reconstruction of areas and relative distances in the loss function of spatial relation decoder. Furthermore, we conducted additional ablation experiments by removing visual features to evaluate the effectiveness of the spatial features extracted by our model. The experiments are performed on PascalVOC dataset. As shown in Table 5, compared with the baseline, the results demonstrate that all modules bring substantial performance gains, the reconstruction of relative distance is more important in the decoder, and the spatial relation decoder contributes most to the overall performance. Moreover, with the fact that only taking spatial features as node features can achieve relatively good performance, it further proves the effectiveness of the spatial features learned from our PR-EnDec.

We also conduct parameter analysis to select hyperparameters. As shown in Table 6, PREGM achieves the best performance when learning rate = $9 \times 10^{-4}$, which is our default setting, and it also shows that adjusting the learning rate causes an accuracy fluctuation of about 1%. For the balance factor $\lambda$ in the loss function, when $\lambda = 1/128, 1/32$, and $1/8$, the matching accuracy is 83.7%, 84.1%, and 83.3% respectively. Thus we select $\lambda = 1/32$ as our default setting, and the results show that our method is rather sensitive to the choice of $\lambda$, and our designed loss function in positional encoder and spatial relation decoder indeed improves the performance.

## 6 Conclusion

In this paper, we present an end-to-end novel deep graph matching network PREGM that learns more distinguishable node features by modeling spatial positional information in graphs. Our PREGM designs a common encoder-decoder architecture consisting of a positional encoder that learns effective node positional encoding and a spatial relation decoder that reconstructs the positional structure of graphs. Our experiments and ablation studies on four public keypoint matching datasets demonstrate the state-of-the-art performance of our method. The exploration direction of future work includes optimizing the loss function of the positional encoder to extract purer spatial structural information and improve the feature fusion method to obtain better fused feature representation.

## Acknowledgement

We acknowledge support from National Key R&D Program of China (2021ZD01133001).

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
