# OpenReview forum: "Improving Graph Matching with Positional Reconstruction Encoder-Decoder Network"
_NeurIPS.cc/2023/Conference — NeurIPS 2023 poster_

### Official Review · Reviewer_1xTE · 2023-06-28

**Soundness:** 4 excellent
**Presentation:** 4 excellent
**Contribution:** 3 good
**Rating:** 6
**Confidence:** 4

**Summary:**

In this paper, the authors propose to improve the existing deep graph matching paradigm via positional reconstruction. The authors claim that existing deep GM works mainly focus on the visual feature to compute the affinities while neglecting the location and position of the key points in the graphs. To this end, they propose to capture the spatial features as well as the visual features in their model PREGM. The proposed encoder decoder PR-EnDec can learn effective graph positional encoding/coding with well-designed loss functions. The experiments on multiple datasets show the performance of the proposed method.

**Strengths:**

1. The consideration of utilizing spatial information is interesting to me since existing deep GM works mainly focus on how to improve the extraction of visual features. The positional encoding is further fused with the visual features, and updated by the well-defined loss functions.
2. The experiments on multiple datasets show that the proposed method PREGM can outperform current SOTA methods.
3. The paper is well-written and easy to follow.


**Weaknesses:**

1. In fact, in the framework of existing works such as BBGM, the geometry information is already the input of SplineCNN. So, I think the authors may over-claim their novelty since existing works also consider spatial information to some extent.
2. In the experiments on Willow and Spair71k datasets, not all baselines are reported in the table. I wonder is there a reason for that? I check the best baseline (COMMON) in the Pascal VOC dataset and find that they have reported their performance on Spair71k (84.5\%), which is higher than the performance of the proposed method PREGM.

**Questions:**

1. I think the authors should further show the difference in spatial information between their work and BBGM since BBGM does consider spatial information.
2. The experiments table may need to be completed. I think it is OK when your method does not outperform all the baselines in one of the datasets, but you need to show the results, right?

---

> ### Author Rebuttal · Authors · 2023-08-09
>
> We deeply appreciate your thorough evaluation and insightful comments on our paper. Your feedback has significantly contributed to refining the clarity and impact of our work.
>
> We are committed to addressing your queries and suggestions:
>
> Claim of Novelty Regarding Spatial Information: We sincerely acknowledge your perspective and understand that the framework of existing works, such as BBGM, incorporates spatial information to a certain extent. In our manuscript, we aim to highlight that we are further applying spatial information in a distinct manner. We appreciate your guidance in framing our contribution accurately.
>
> Difference in Spatial Information:
> We thank the reviewer for suggesting a deeper analysis of the distinction in spatial information between our work and BBGM. Our method introduces spatial information at multiple levels:
>
> a. Higher-order Information: The parameters involved in the splineCNN training within BBGM do not encompass spatial information beyond order 2 (edges). Our method, on the other hand, extends this by incorporating spatial information directly into the positional encoding of individual nodes.
>
> b. Global Information: Our encoder directly facilitates information exchange between nodes, bypassing the reliance on edge construction through triangular dissection. This design choice enables better global spatial information utilization compared to BBGM.
>
> As the reviewer pointed out, both approaches harness spatial information, but the embodiment and integration of this information differ significantly. We will elucidate these distinctions further in the revised manuscript to ensure clarity.
>
> Incomplete Experimental Results:
> We acknowledge the reviewer's concern regarding missing baseline results in the experimental table, particularly for the Willow and SPair71k datasets. We appreciate the suggestion to include all relevant baseline data. The omission of these results was due to the unavailability of some baseline methods' data on these specific datasets.
>
> Specifically, the baseline method COMMON exhibits an accuracy of 84.5\% on the SPair71k dataset, surpassing our proposed method. However, we observed a discrepancy in the reported accuracy of BBGM across different experimental sources(78.9\% in the original paper, 82.1\% in COMMON's experimental data), which makes it challenging to directly compare results. Moreover, differences in dataset parameters may contribute to these variations. We have taken the reviewer's advice seriously and tested the open-source code of COMMON with the same settings and achieved an accuracy of 82.85\% on the Spair71k dataset. We will include this result in the revised manuscript..
>
> Furthermore, we apologize for any confusion caused by not fully populating the experimental table. We commit to diligently addressing this concern by providing comprehensive experimental results, including those where our method does not achieve the best performance. We appreciate the reviewer's understanding and will ensure that our revised manuscript reflects these improvements.
>
> Your hints have been immensely valuable in providing context and guiding our responses. Your dedication to a thorough evaluation has motivated us to enhance the rigor and clarity of our research.
>
> Once again, we express our gratitude for your diligence and insightful feedback.

---

> > ### Comment · Reviewer_1xTE · 2023-08-16
> >
> > Thank you for your detailed response. My doubts have been cleared.
> >
> > However, I take a look at other reviewers' comments and find one thing I care about:
> >
> > In your reply to Reviewer LuNk, you claimed that "We want to clarify that the convention of comparing graphs with equal numbers of points is commonly used in graph matching.", which I do not think so. In many existing deep GM works, such as NGM and BBGM, the number of points in two images is not required to be equal. In the BBGM paper, they use two filtering to handle the cases when two images do not have the same points, namely Intersection filtering and Inclusion filtering, shown in Figure 6 of the BBGM paper. Therefore, I do not think the "Keypoints Number Constraint" claim is correct.

---

> > > ### Author Response · Authors · 2023-08-17
> > >
> > > Thank you for your thoughtful comment. We appreciate your engagement with our response to Reviewer LuNk's concerns. Your feedback has provided us with valuable perspectives that will guide our research in the future. We would like to clarify our stance on the matter. While it's true that certain deep graph matching works like NGM and BBGM do not strictly require an equal number of points in the compared graphs, we also want to emphasize that the convention of comparing graphs with equal numbers of points is indeed prevalent in the field of graph matching.
> > >
> > > We acknowledge the inclusion filtering method employed in the BBGM paper to handle cases where two images do not have the same points. Non-equal point matching is indeed valuable for addressing more flexible matching scenarios. However, our claim about the "Keypoints Number Constraint" was intended to highlight a common practice in the field and was not meant to discount alternative approaches.
> > >
> > > We also agree that exploring non-equal point matching is an interesting avenue for future research. As you mentioned, it's clear that the BBGM and NGM papers have demonstrated successful techniques for addressing this challenge. Moving forward, we plan to investigate such approaches as part of our ongoing efforts to enhance the robustness and versatility of graph matching techniques.

---

> > > > ### Comment · Reviewer_1xTE · 2023-08-18
> > > >
> > > > Thanks for the response. For now, I will keep my score weak accept.

---

### Official Review · Reviewer_LuNk · 2023-07-01

**Soundness:** 3 good
**Presentation:** 3 good
**Contribution:** 3 good
**Rating:** 6
**Confidence:** 4

**Summary:**

This paper introduces a positional reconstruction encoder-decoder (PR-EnDec) to model intrinsic graph spatial structure for image key-points matching. By using graph to represent the image structure, the proposed model can utilize the high-order spatial information by reconstructing  the locational structure of graphs contained in the node coordinates.

**Strengths:**

1.The structure of the whole manuscript is good. The introduced positional encoder that learns effective graph positional encoding with affine transformation invariance.

2. The figures and tables are in good illustration.

**Weaknesses:**

1.For experimental comparison, the compared methods are old. Besides, the evaluation datasets, such as PascalVOC [8], Willow ObjectClass [4], and SPair-71k  are small and little old.  Some more recent large-scale datasets, such as PhotoTourism (IMC-PT) 2020 should be evaluated.

2.Given the Problem Formulation of Graph Matching in sec.3, the compared two graphs have the same number of nodes. However, for the real application, the keypoints number constraint is not reasonable.

3.Figure 2 shows the framework of the proposed PR-EnDec network that consists of the detailed structure of its positional encoder and spatial relation decoder.  However, in figure1, the Spatial Relation Decoder is missing.

4.Compared to the previous works, the two stage training pipeline is complex.

**Questions:**

The proposed method takes the reference [28] as the baseline model (line 278), thus, this method should be compared on all three datasets.

**Limitations:**

See weakness and questions

---

> ### Author Rebuttal · Authors · 2023-08-09
>
> We extend our sincere appreciation for your diligent review and insightful feedback on our paper. Your thoughtful comments have greatly contributed to the enhancement of our work.
>
> We are committed to addressing your concerns and queries:
>
> Experimental Comparison and Dataset Choice: We thank you for highlighting the importance of using recent and large-scale datasets for evaluation. In response, we have included experimental results on the PhotoTourism (IMC-PT) 2020 dataset in the Appendix. Furthermore, we will present comprehensive tabular data and in-depth analysis of the results in the revised manuscript to ensure a thorough understanding of the proposed method's performance.
>
> Keypoints Number Constraint in Problem Formulation: We want to clarify that the convention of comparing graphs with equal numbers of points is commonly used in graph matching. However, we acknowledge the importance of considering scenarios with unequal keypoints, as it aligns more closely with real-world applications. In line with your suggestion, we are planning to explore and incorporate unequal point matching as a valuable future direction for our research.
>
> Figure 1 and Spatial Relation Decoder: We appreciate your attention to the spatial relation decoder. While it might appear missing in Figure 1(graph matching phase), please note that it is not absent from the overall framework. The decoder is employed specifically to enhance the training of the positional encoder. Its primary purpose is to improve the performance of the positional encoder by providing a reconstruction target during the training phase.
>
> Complexity of Two-Stage Training Pipeline: Your observation about the complexity of the two-stage training pipeline is accurate. We acknowledge that the concurrent training of positional encoders and graph matching presents a certain level of complexity. In light of this, we'd like to highlight that even the approach DGMC employed a two-stage training pipeline. While our results demonstrate superior performance, we acknowledge your point. Moving forward, we envision training positional encoders and conducting graph matching simultaneously as a promising direction.
>
> Comparison with Baseline Model: We appreciate your observation and confirm that we have compared the proposed method with the baseline model BBGM on all three datasets.
>
> Once again, we are genuinely grateful for your feedback, and we are committed to addressing these issues to ensure the clarity, comprehensiveness, and effectiveness of our paper. Your dedication to fostering high-quality research and constructive critique is invaluable.
>
> Thank you for your time and consideration.

---

> > ### Comment · Reviewer_LuNk · 2023-08-21
> > **About the rebuttal**
> >
> > Thanks for the providing results. The rebuttal has answered all my concerns. I raise the core to weak accept.

---

### Official Review · Reviewer_TCc6 · 2023-07-06

**Soundness:** 3 good
**Presentation:** 2 fair
**Contribution:** 3 good
**Rating:** 6
**Confidence:** 5

**Summary:**

The paper introduces an improved method for graph matching in semantic keypoint matching - the Positional Reconstruction Encoder-Decoder Network (PR-EnDec) and an end-to-end graph matching network PREGM. PR-EnDec efficiently learns node spatial embedding and reconstructs the locational structure of graphs from node coordinates. PREGM models the intrinsic spatial structure of keypoints and captures visual information, enhancing node positional features. Tests on three keypoint matching datasets showed improved performance over existing methods, with an ablation study demonstrating the effectiveness of each PREGM component.


**Strengths:**

The paper introduces an innovative PR-EnDec model to effectively capture and utilize the spatial context information hidden in the locations of keypoints, which has not been adequately addressed in existing methods. The PR-EnDec incorporates a positional encoder and a spatial relation decoder, which not only capture the relative spatial relations but also learn the affine transformation invariance, enabling the network to learn more refined location information. Given the widespread use of image matching in various fields, such as object tracking, image retrieval, and pose estimation, the proposed improvements could have significant impacts on a variety of applications.

**Weaknesses:**

1. Lack of Visualizations

While the paper presents an innovative approach and comprehensively assesses the performance of the proposed method, I would like to express a concern regarding the absence of sufficient visualizations. Visualizations can offer empirical evidence of model performance and further support the reported quantitative results. For instance, visual representations of keypoint matching results can provide intuitive insights into how the proposed method works in practice and highlight its ability to handle complex spatial relationships. In light of the above, the lack of sufficient visualizations in this paper might impede comprehensive understanding and thorough evaluation of the presented method. Therefore, I would suggest incorporating necessary visual aids into the paper to provide a more effective and comprehensive presentation of the methodology, the performance, and the practical implications of the proposed PREGM model.

2. The Formatting of Table 4 and Table 5

In the current presentation, multiple experimental results appear to be consolidated into single rows within these tables. This presentation may potentially lead to confusion and misinterpretation of the results. Placing more than one set of results on a single line may obscure important details, making it harder for readers to draw meaningful conclusions from the data. As a suggestion to enhance clarity and readability, it would be beneficial to dedicate each row to one specific experimental result. This layout would allow for more detailed descriptions of the corresponding experimental settings and the related results, thus improving understanding. Moreover, it will facilitate the direct comparison of different experimental conditions and results, which is particularly critical in identifying trends, nuances, and potential implications. Therefore, I would recommend revising the formatting of Table 4 and Table 5 to present one set of experimental results per row, thereby ensuring that the wealth of information is conveyed as comprehensibly as possible.

3. Lack of discussion about limitations and potential negative societal impact

While the paper exhibits a commendable effort in proposing and testing a new methodology in graph matching, I would like to raise a concern regarding the lack of discussion on the potential limitations of the proposed method and its possible negative social implications. I suggest that the authors include a discussion of the possible limitations of the PREGM model and potential negative societal impacts in the paper, thereby presenting a more comprehensive and nuanced understanding of their work.

**Questions:**

Please see the weaknesses part in the above section.

**Limitations:**

Limitations are not described in this paper, and potential negative societal impact is missing.

---

> ### Author Rebuttal · Authors · 2023-08-09
>
> We extend our gratitude for your meticulous evaluation and constructive feedback on our paper. Your insights have significantly contributed to the refinement of our work.
>
> We are committed to addressing your concerns and queries:
>
> Lack of Visualizations: We sincerely appreciate your suggestion regarding the inclusion of visualizations. Visual aids indeed offer a valuable means of providing empirical evidence and enhancing the clarity of our proposed methodology and its performance. We acknowledge the importance of intuitive insights, and we will ensure the incorporation of meaningful visual representations of keypoint matching results in our revised manuscript. These visualizations will help elucidate the inner workings of our approach and provide readers with a comprehensive understanding of its practical applications.
>
> The Formatting of Table 4 and Table 5: Your feedback regarding the formatting of tables is duly noted. We understand the significance of clear and concise presentation for facilitating an accurate interpretation of experimental results. In line with your recommendation, we will reformat Table 4 and Table 5 to ensure that each set of experimental results is presented in a dedicated row. This adjustment will improve the clarity and readability of the tables, enabling readers to readily compare different experimental conditions and outcomes.
>
> Lack of Discussion about Limitations and Potential Negative Societal Impact: We appreciate your concern regarding the omission of a discussion on the limitations of our proposed method and its potential societal implications. Specifically, equal point matching is acknowledged as one of our limitations in our work. In the revised version of our paper, we will thoroughly address the possible limitations of the PREGM model and consider potential negative societal impacts associated with its application.
>
> We genuinely value your feedback and suggestions, as they underscore our commitment to enhancing the quality and comprehensiveness of our research. Your dedication to fostering a deeper understanding of our contributions is invaluable.
>
> Thank you once again for your time and insightful review.

---

> > ### Comment · Reviewer_TCc6 · 2023-08-19
> >
> > After thoroughly reviewing the authors' rebuttal and considering the feedback from other reviewers, I appreciate the detailed responses provided to address the concerns raised. The commitment to incorporate visualizations, reformat the tables for clarity, and the addition of a discussion on the potential limitations and societal implications of the PREGM model are commendable. These revisions promise a more comprehensive presentation of the paper. Based on these considerations, I maintain my initial rating of "weak accept" for this submission.

---

### Official Review · Reviewer_pkGJ · 2023-07-07

**Soundness:** 3 good
**Presentation:** 3 good
**Contribution:** 3 good
**Rating:** 6
**Confidence:** 4

**Summary:**

This paper presents a new method to improve graph matching by supplementing visual features with positional encodings. Specifically, an encoder-decoder model is pre-trained to reconstruct a graph’s relative spatial relation based on node coordinates only. The encoder is additionally trained under a contrastive loss to classify positive or negative graph pairs. The pre-trained encoder provides the positional encodings, which is used to compute node and edge affinities with CNN visual features jointly. The proposed method is validated on three graph matching datasets: PascalVOC, WillowObject Class, and Spair-71k, with the best performance among all compared methods. Ablation study shows the contribution of different components in the proposed method.


**Strengths:**

1. The proposed positional reconstruction encoder-decoder network is a simple and effective method to extract positional features. Reconstructing the spatial relation between nodes in a graph is a good target for the encoder-decoder network.
2. This paper proposes to use positional encodings to improve graph matching, which proves to be effective on three datasets. This is also in line with conclusions of some existing research, such as the vision transformer, in which positional encodings are shown to be very important. This work suggests all future work on graph matching can benefit by adding positional features.
3. The ablation study has shown the importance of each component in the proposed method, including both encoder and decoder, the reconstruct targets, and visual features. Parameter analysis also provides the reason for the choice of hyper-parameters.
4. This paper is well-written and easy to follow.


**Weaknesses:**

Some technical details of the proposed method are not clarified. What are the detailed configurations of the multi-head self-attention layers? How are positive and negative pairs for the encoder generated? Are the visual features sampled for each node of the graph? How are the graph edges defined? How are the edges used in the encoder? How are the node and edge affinities computed?

**Questions:**

1. For the visual features, why is VGG-16 used instead of a deeper model like ResNet-50?
2. For the decoder reconstruction target, is the original coordinate, which is a natural choice, a good choice? If not, why?


**Limitations:**

This paper has not discussed the limitations.

---

> ### Author Rebuttal · Authors · 2023-08-09
>
> We sincerely appreciate your thorough review and valuable feedback on our paper. Your insightful comments have greatly contributed to the refinement and clarity of our work.
>
> We are pleased to address your specific concerns and queries:
>
> Detailed Configurations of Multi-Head Self-Attention Layers: For each multi-head self-attention layer, we employed an attention mechanism with 8 attention heads and a feed-forward network. This configuration allows the model to capture complex relationships between nodes effectively.
>
> Generation of Positive and Negative Pairs for the Encoder: Positive pairs are identified when graph $\mathcal{G}^2$ is affine transformed by graph $\mathcal{G}^1$, or keypoints in both graphs are one-to-one matched. Negative pairs correspond to situations where keypoints in two graphs are shuffled and not one-to-one matched. We expect the corresponding node positional encoding $f_i^1$ and $f_i^2$ should be similar only if node $V_i^1$ matches $V_i^2$.
>
> Sampling Visual Features for Each Node of the Graph: Yes, we sampled visual features for each node. We computed feature vectors using the relu4\_2 and relu5\_1$ layers of the VGG16 network. These feature vectors are spatially interpolated to correspond to keypoints using bi-linear interpolation.
>
> Definition and Use of Graph Edges in the Encoder: The edges within each graph are generated through Delaunay triangulation. However, edge information is not incorporated into the position encoder. Instead, it plays a key role in the message passing module during the graph matching phase.
>
> Computation of Node and Edge Affinities: Node affinities are calculated using a similar approach to the
> unary costs in BBGM. Specifically, we compute $c^v_{i,j}=\sum_kf^v_s(i)_ka_kf^v_t(j)_k$, where $f^v_s(i)$ and $f^v_t(j)$ represent feature vectors for vertices $i$ and $j$ in the source and target graphs, respectively. The affinity $a$ is obtained using a one-layer neural network from the global feature vector $g$ extracted by max-pooling the final VGG16 layer. Edge affinities, on the other hand, are not employed in our model.
>
> Choice of VGG-16 over Deeper Models: We used VGG-16 as a convention inherited from PCA. While we experimented with replacing it with ResNet-50, the improvement was not significant, leading us to stick with VGG-16 for consistency.
>
> Decoder Reconstruction Target: The choice of using original coordinates as the reconstruction target has its rationale. While the translation invariance is somewhat compromised, we focus on capturing high-order spatial relationships between nodes. First-order spatial information as a target would reduce the encoder's utilization of spatial information, diminishing the model's overall performance.
>
> Once again, we express our gratitude for your insightful feedback. Your constructive critique has helped us enhance the clarity and effectiveness of our proposed method. We are committed to addressing the remaining technical details and ensuring a comprehensive discussion of limitations in the revised manuscript.
>
> Thank you for your dedication to improving the quality of scientific research.

---

### Official Review · Reviewer_Pb59 · 2023-07-26

**Soundness:** 4 excellent
**Presentation:** 4 excellent
**Contribution:** 4 excellent
**Rating:** 7
**Confidence:** 1

**Summary:**

The authors introduce a positional reconstruction encoder-decoder (PR-EnDec) to model intrinsic graph spatial structure, and present an end-to-end graph matching network PREGM based on PR12 EnDec. The PR-EnDec consists of a positional encoder that learns effective node spatial embedding with the affine transformation invariance, and a spatial relation decoder that further utilizes the high-order spatial information by reconstructing the locational structure of graphs contained in the node coordinates.

**Strengths:**

I haven not personally conducted any research on graph matching, but the idea look interesting to me.
I suggest AC rely on other more experienced reviewer in this area.

**Weaknesses:**

My knowledge of graph matching is limited and therefore cannot find any weakness.

**Questions:**

What is the purpose of graph matching? Any use case related to NeRFs or generative models?


**Limitations:**

I have no idea about the limitations of graph matching.

---

> ### Author Rebuttal · Authors · 2023-08-09
>
> We sincerely appreciate your time and effort in reviewing our paper on graph matching. Your feedback is invaluable to us as it provides insights that contribute to the overall quality of our work.
>
> To address your question about the purpose of graph matching, we are glad to provide a brief explanation. Graph matching plays a crucial role in various fields, including computer vision, image analysis, and pattern recognition. Its purpose is to establish correspondences between nodes of different graphs, facilitating tasks such as object recognition, image alignment, and shape matching.
>
> Once again, thank you for your time, and we look forward to incorporating your suggestions and improving our work based on your feedback.

---

### Author Rebuttal · Authors · 2023-08-09

The attachment contains the experimental results for IMC-PT.

---

### Decision · Program_Chairs · 2023-09-21

**Decision:**

Accept (poster)

**Comment:**

All reviewers unanimously recommended acceptance, appreciating the novel idea of positional encoding for graph matching as well as strong experimental results. There were concerns about missing details and comparisons, and the rebuttal addressed most of them, satisfying the reviewers. AC also agrees with the reviewers’ consensus that this paper provides an interesting contribution to the community, recommending acceptance. AC encourages the authors to incorporate the reviewers’ comments into the final version.